# The possibility of integrating motile sperm organelle morphology examination (MSOME) with intracytoplasmic morphologically-selected sperm injection (IMSI) when treating couples with unexplained infertility

**Aula Asali**[1]*, **Netanella Miller**[1], **Yael Pasternak**[1], **Vita Freger**[2], **Michael Belenky**[2], **Arie Berkovitz**[1,3,4]

1 Department of Obstetrics and Gynecology, Meir Medical Center, Kfar Saba, Israel, 2 Male Fertility Center, Rishon LeZion, Israel, 3 Sackler School of Medicine, Tel Aviv University, Tel Aviv, Israel, 4 Assuta Medical Center, Tel Aviv, Israel

* aula_atamna@yahoo.com

## Abstract

### Purpose

To examine the efficacy of motile sperm organelle morphology examination (MSOME) and intracytoplasmic morphologically-selected sperm injection (IMSI) for unexplained infertility.

### Methods

This historical study, included 271 couples with primary, unexplained infertility/male subfertility, treated at an outpatient, IVF clinic, 2015–2018. These couples underwent MSOME after ≥3 failed intrauterine insemination (IUI) cycles and ≥1 failed IVF-ICSI cycle. They proceeded to intracytoplasmic morphologically-selected sperm injection (IMSI) within 6 months of MSOME. IMSI is conducted on the day of oocyte pick-up with a fresh semen sample. Pregnancy and delivery rates were analyzed.

### Results

The cohort was divided based on percentage of normal cells at MSOME: Group A included 55 with no normal cells, Group B, 184 with 0.5%≤ normal cells ≤1.5% and Group C, 32 with ≥2% normal cells. Normal spermatozoa were found in 49 (89%) of Group A after extensive search. Group A had higher pregnancy rate (62.7%) compared to B (47.2%, $P = 0.05$) and C (28.1%, $P = 0.002$). Group B had higher pregnancy rate than C (p = 0.045). Delivery rate was higher in Group A (52.1%) compared to B (34.1%, p = 0.023) and C (21.9%, p = 0.007). Pregnancy and delivery rates were higher in A compared to B+C (p = 0.018, p = 0.01, respectively).

### Conclusions

MSOME may be useful for evaluating unexplained infertility. IMSI can be recommended for men with <2% normal spermatozoa at MSOME.

**Data Availability Statement:** All relevant data are within the paper and its Supporting Information files.

**Funding:** The authors received no specific funding for this work

**Competing interests:** Dr. Berkovitz is the Medical Director of the Male Fertility Clinic where the MSOME was performed. The other authors have no conflicts to declare. This does not alter our adherence to PLOS ONE policies on sharing data and materials.

## Introduction

Unexplained infertility refers to the absence of a definable cause for a couple's failure to conceive after 12 months despite thorough evaluation, or after 6 months among women 35 and older [1]. It affects 10–30% of infertile/sub-fertile couples [2,3].

In contrast to clearly diagnosed conditions, such as ovulatory disorders, unexplained infertility may encompass situations that are overlooked by conventional infertility assessment, or are undetectable using current diagnostic techniques. A systematic review concerning treatment options for couples with unexplained infertility concluded that current data were not compelling enough to indicate a specific treatment modality that would significantly increase the chance of conception and suggested that these couples should be assessed and treated individually [2,4].

Motile sperm organelle morphology examination (MSOME) is performed in real-time using an inverted light microscope equipped with high-power Nomarski optics, enhanced with digital imaging to achieve magnification up to 6300× [5]. Thus, MSOME enables detecting subtle sperm organellar malformations in motile spermatozoa that an embryologist might consider normal for fertilization at 200× to 400× magnification [5]. Using this technique together with a micromanipulation system has allowed the introduction of a modified intracytoplasmic sperm injection (ICSI) procedure, known as intracytoplasmic morphologically-selected sperm injection (IMSI) [6]. When the spermatozoa with normal morphology and motility selected for ICSI are detected under a magnification of ×400, in IMSI the motile spermatozoa are selected under magnification up to 6300×[5].

To the best of our knowledge, there are no data regarding the role of MSOME and possibly IMSI in evaluating couples with unexplained infertility. This study examined the efficacy of IMSI in unexplained infertility when conception did not occur after at least one IVF-ICSI cycle. We examined also the possible benefit of IMSI in relation to MSOME results.

## Materials and methods

This historical study included 271 couples. The inclusion criteria were couples with primary unexplained infertility/male subfertility who had experienced at least three failed IUI cycles and at least one failed IVF-ICSI cycle, from 2015 through 2018 in Assuta Medical Center, Rishon LeZion, Israel. All couples underwent IMSI within 6 months of the initial MSOME. IMSI is conducted on the day of oocyte pick-up with a fresh semen sample.

Since 2015, when MSOME was incorporated in our lab, we offer MSOME evaluation to couples with unexplained infertility who had at least one failed IVF-ICSI cycle. Male subfertility was defined as a total motile sperm count $\geq 1 \times 10^6$ [7].

Data were collected regarding the first IVF-IMSI cycle after examination. Information regarding transferred frozen embryos of IVF-IMSI cycles (cumulative pregnancy rate) were included, as well.

Exclusion criteria were Mullerian abnormality or submucosal or subserosal fibroids >7 cm, despite regular menstrual cycles, because of the negative effect of these complications on implantation and pregnancy outcomes [8]; ovulation abnormalities; tubal pathology; and male infertility (when the total motile sperm count was $<1 \times 10^6$, because IUI treatment does not benefit this group).

Data concerning routine semen analysis, percentage of normal cells at MSOME, basic demographic characteristics of the female partner, number of retrieved oocytes, number of transferred embryos, day of transfer and pregnancy outcomes (abortion rate, delivery rate) were collected from electronic medical records.

Clinical pregnancy, defined as missed abortion, blighted ovum or fetal pole with heartbeat, was considered a positive result. Chemical pregnancy (when the level of hCG is initially elevated enough to produce a positive result on a pregnancy test but then declines before a gestational sac is detected with ultrasound) was considered a negative result.

The cohort was divided into 3 groups according to the percentage of normal cells at MSOME. Group A included 55 cases with no normal cells at MSOME. Group B included 184 cases with 0.5% to 1.5% normal cells at MSOME and Group C consisted of 32 cases of at least 2% normal cells at MSOME. This reflects the normal MSOME, as a fertile male has an average of 2% normal cells (laboratory data, unpublished).

The primary outcomes of cumulative pregnancy rate and delivery rate were compared between the three subgroups of the cohort (A, B and C), in order to define the population who may benefit most from IMSI.

## Spermatozoa collection and processing for MSOME

Sperm samples were obtained by masturbation or by using a spermicide-free polyurethane condom. After being allowed to liquefy at room temperature for 20 minutes, the samples were loaded on PureCeption Sperm Separation Media (Sage In-Vitro Fertilization, Inc. Trumbull, CT, USA) gradient of 1 mL 40% v/v (upper phase) and 1 ml 80% v/v (lower phase). This consisted of a sterile colloidal suspension of silane-coated silica particles in HEPES-buffered human tubal fluid containing 10 mg/l gentamicin. It was centrifuged for 20 minutes at 300 relative centrifugal field at room temperature. The upper liquid was then removed and the pellet was re-suspended in Quinn's Sperm Washing Medium (Sage In-Vitro Fertilization, Inc.) and centrifuged again for 5 minutes at 600 rpm at room temperature. After this procedure was performed twice, the pellet was re-suspended in sperm washing medium at a spermatozoa concentration of approximately $10^7$/ml. A 4 μl droplet of polyvinyl pyrrolidone (PVP) 10% solution (Sage In-Vitro Fertilization, Inc.) was placed in a glass-bottom tissue culture dish (World Precision Instruments, Sarasota, FL, USA) and covered with paraffin oil (Sage In-Vitro Fertilization, Inc.). Next, 1 μl of the sample suspension was loaded onto the PVP droplet. The droplet containing the spermatozoa was examined under 6000x magnification using an inverted phase contrast microscope Nikon Eclipse Ti (Nikon Instruments, Inc., Melville, NY, USA) equipped with differentiated interference contrast, and an Invenio 3SII camera. The magnification was achieved by using an oil-covered 100x lens, while the additional 60x magnification was provided by DeltaPix software (Smorum, Denmark).

Then, 200 motile spermatozoa along the borders of the droplet were examined for size, shape and presence of nuclear vacuoles and disorders, Fig 1.

The percentage of normal sperm head shapes was calculated based on the following criteria: length 4.75 ± 0.28 μm and width 3.28 ± 0.20 μm; symmetric oval shape; smooth texture; no more than 2 acrosomal vacuoles comprising <4% of the sperm head area; no post-acrosomal or deep vacuoles; centrally located midpiece and no regional disorders such as cytoplasmatic extrusions or invaginations [9–11].

## Ethics statement

The study was approved by the Assuta Medical Center Ethics Committee. Due to the retrospective nature of the study, informed consent was not required.

Data will be made available upon reasonable request to the corresponding author.

## Statistical analysis

Data are described as mean, standard deviation, minimum and maximum for continuous variables and as percentage of total for nominal parameters. In univariate analyses, Chi-Square test

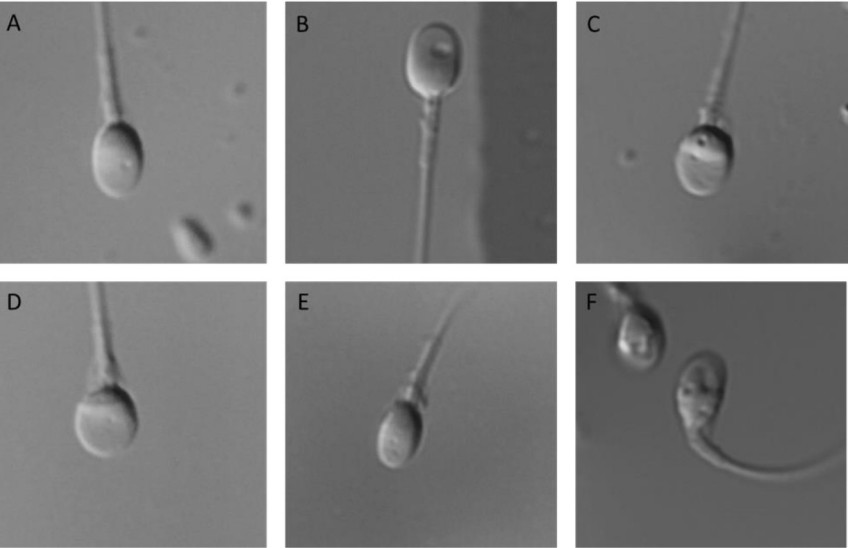

**Fig 1. Sperm head morphology at 6000x magnification.** A. Normal morphology. B. Large acrosomal vacuole. C. Equatorial and post-acrosomal vacuoles. D. Wide head. E. Narrow head. F. Highly vacuolated head.

or Fisher's Exact test were used for non-metric parameters, each when appropriate. One-way analysis of variance or Kruskal-Wallis non-parametric test was used to compare three groups at different levels of MSOME, each by matching the rules. Bonferroni *post hoc* comparisons were used (to adjust *P*-values when several dependent or independent statistical tests were performed simultaneously on a single data set), The alpha used for the Bonferroni Correction was 0.05/3 = 0.017. Logistic regression was used to evaluate the effect of parameters that were statistically different by pregnancy rate in the univariate analysis. The multinomial significance levels were set at 0.05. All statistical analyses were performed using SPSS-25.

## Results

The cohort included 312 couples who underwent MSOME after at least one failed IVF-ICSI attempt. Among them, 68 (21.8%) couples had isolated teratozoospermia (no normal spermatozoa in MSOME), 205 (65.7%) had male subfertility and 39 (12.5%) had normozoospermia. Among the 312 couples, 41 did not complete follow-up or underwent IMSI more than 6 months after the MSOME examination. Thus, 271 proceeded to IMSI within 6 months of the initial MSOME.

The basic characteristics of the cohort are shown in Table 1. The mean age of the male partners was 37.6 ± 5.8 years. The female partners were a mean age of 35.5 ± 5.5 years.

Results of the logistic regression on pregnancy rate are shown in Table 2. Older female age and higher percentage of normal cells at MSOME had negative effects on pregnancy (p = 0.001 and p = 0.006, respectively). The day of embryo transfer had a positive effect on pregnancy (p = 0.009). No correlations were found between pregnancy and age of male partner (p = 0.101), BMI of female partner (p = 0.817), number of oocytes retrieved (p = 0.173) or number of transferred embryos (p = 0.1).

Cleaved embryos were transferred in 236 cases (91.8%) and blastocysts were transferred in 21 (8.2%). The cumulative pregnancy rate of the entire cohort was 46.9% (126 couples conceived after the first IVF-IMSI cycle). Four couples (3.2% of pregnancies) had multiple gestations.

**Table 1. Basic characteristics of the cohort.**

| Characteristic | Mean ± SD | Minimum | Maximum |
|---|---|---|---|
| Age of male partner, years | 37.6 ± 5.8 | 24 | 60 |
| Age of female partner, years | 35.5 ± 5.5 | 23 | 45 |
| BMI of female partner | 23.35 ± 4.12 | 16 | 37 |
| Volume of the sample, ml | 2.78 ± 1.5 | 0.2 | 9 |
| Concentration, $10^6$/ml | 57.28 ± 47.7 | 0 | 250 |
| Total motile count | 75.92 ± 85.26 | 0.15 | 501.9 |
| Motility, % | 45.98 ± 18.54 | 3 | 93 |
| Percentage of normal cells at MSOME | 1.32 ± 1.5 | 0 | 8 |
| Number of oocytes picked up | 7.92 ± 4.5 | 1 | 24 |
| Number of transferred embryos | 2.13 ± 0.717 | 1 | 4 |

## Comparison between the three subgroups of the cohort

The cohort was grouped according to the percentage of normal cells at MSOME. Table 3 shows the demographic and clinical characteristics of the 3 groups. Group A included 55 cases in which no normal cells were found at MSOME, but normal cells were found during IMSI at ovum pick-up for 49 (89.1%). Group B included 184 couples and Group C 32 couples.

A significant difference was found between the 3 groups regarding cumulative pregnancy rate (p = 0.008) and delivery rate (p = 0.015). We found higher pregnancy rate in Group A (62.7%) compared to Group B (47.2%, p = 0.05) and Group C (28.1%, p = 0.002) and in Group B compared to C (47.2% vs. 28.1%, p = 0.045). Higher delivery rate was found in Group A (52.1%) compared to B (34.1%, p = 0.023) and C (21.9%, p = 0.007). No differences in delivery rates were found between Groups B and C (34.1% vs. 21.9%, respectively, p = 0.174).

Total motile count differed between groups (p = 0.003). It was lower in group A than in Group B (p = 0.005) and Group C (p = 0.011).

No difference was found regarding the age of the male partner (p = 0.283), the female partner (p = 0.264) or BMI (p = 0.447). No difference was found between groups in sample volume (p = 0.131), motility of the spermatozoa (p = 0.785), number of retrieved oocytes (p = 0.06), number of transferred embryos (p = 0.176), transfer day (p = 0.16) or abortion rates (8.3% vs. 11% vs. 6.3%, p = 0.657).

**Table 2. Logistic regression on pregnancy rate.**

| Parameter | Significance | Odds ratio | 95% CI for OR | |
|---|---|---|---|---|
| | | | Lower | Upper |
| Male partner age, years | 0.101 | 1.080 | 0.985 | 1.185 |
| Female partner age, years | 0.001* | 0.840 | 0.758 | 0.931 |
| BMI of female | 0.817 | 1.008 | 0.940 | 1.082 |
| Percentage of normal cells at MSOME | 0.006* | 0.374 | 0.186 | 0.754 |
| Number of oocytes picked up | 0.173 | 1.066 | 0.972 | 1.168 |
| Number of transferred embryos | 0.1 | 1.438 | 0.933 | 2.215 |
| Day of transfer | 0.009* | 1.929 | 1.178 | 3.158 |

*Significant difference.

CI, confidence interval, OR, odds ratio.

**Table 3. Demographic and clinical characteristics of the three sub-groups.**

| Variable | Group A (n = 55) | Group B (n = 184) | Group C (n = 32) | P-value |
|---|---|---|---|---|
| Female partner age, years | 36.1 ± 5.4 | 35.5 ± 5.6 | 34.1 ± 5.2 | 0.264 |
| Male partner age, years | 38.3 ± 4.9 | 37.6 ± 5.9 | 36.3 ± 6.2 | 0.283 |
| BMI of female partner | 24 ± 4.5 | 23.3 ± 3.98 | 22.8 ± 4.24 | 0.447 |
| Sample volume, ml | 2.72 ± 1.89 | 2.9 ± 1.5 | 2.3 ± 1 | 0.131 |
| % motility | 44.4 ± 18.8 | 46.3 ± 17.9 | 46.7 ± 21.9 | 0.785 |
| Total motile count | 41.8 ± 51.0 | 82.6 ± 85.2 | 96.3 ± 114.98 | 0.003* |
| Number of oocytes picked up | 9.3 ± 4.7 | 7.65 ± 4.4 | 7.19 ± 3.99 | 0.06 |
| Number of transferred embryos | 2.26 ± 0.7 | 2.12 ± 0.7 | 1.97 ± 0.7 | 0.176 |
| Day of transfer | 2.98 ± 0.5 | 3.1 ± 0.6 | 3.25 ± 0.7 | 0.16 |
| Pregnancy rate | 62.7% | 47.2% | 28.1% | 0.008** |
| Abortion rate | 8.3% | 11% | 6.3% | 0.657 |
| Delivery rate (includes ongoing pregnancy) | 52.1% | 34.1% | 21.9% | 0.015* |

*Significant difference between Group A and B, and Group A and C.

**Significant difference between the three groups.

### Comparison between subgroup A and the other study groups

When comparing pregnancy rates between Group A and the rest of the cohort (Groups B and C together), Group A experienced significantly higher pregnancy and delivery rates compared to B+C (62.7% vs. 44.3%, p = 0.018 and 52.1% vs. 32.2%, p = 0.01), respectively. In addition, in Group A more oocytes were picked-up compared to the rest of the cohort (9.3 ± 4.7 vs. 7.58 ± 4.4, p = 0.01). Total motile count was higher in group B+C (84.6 ± 90.03) than in Group A (41.84 ± 51.1, p = 0.001). No difference was found regarding abortion rate (8.33% vs. 10.24%, p = 0.69).

## Discussion

The motile sperm organelle morphology examination (MSOME) was introduced by Bartoov, et al. in 2001 [12]. This examination is based on morphological analysis of isolated motile spermatozoa in real-time, at high magnification (up to 6300×) [13]. **Thus, MSOME can detect subtle organellar malformations in motile spermatozoa that an embryologist might consider normal for fertilization at 200× to 400× magnification [5], especially when a normal sperm nucleus is a significant factor for successful implantation and pregnancy in ICSI procedures [13,14].**

We hypothesized that an unrecognized male factor is involved in the etiology of some cases of unexplained infertility. MSOME, as described above, is more sensitive for detecting subtle sperm head abnormalities. Our cohort included couples who had at least one unsuccessful, conventional IVF-ICSI cycle, assuming that an unknown male factor was not solved in the previous cycle. In support of this assumption, a previous study [15] found an increase in implantation and pregnancy rates using IMSI over conventional ICSI for patients with previously unsuccessful ICSI, with no difference in the results of the first cycle with ICSI vs. IMSI.

To the best of our knowledge, this is the first study to evaluate the utility of MSOME and possible IMSI for unexplained infertility. We found that the percentage of normal cells at MSOME affects pregnancy rates. With IMSI, the lower the percentage of normal cells, the higher the pregnancy rate. An explanation for this apparent contradiction is that a lower percentage of normal cells in MSOME necessitates an extensive search under ultra-magnification to find the highest quality spermatozoa.

This process is not performed for routine procedures, but is performed for IMSI.

Couples with higher percentage of normal cells at MSOME do not benefit from the extensive search; they have enough normal sperm for fertilization.

In addition, men with higher percentage of normal cells in the ejaculate can be managed with routine IVF-ICSI. But, when there are only a few normal spermatozoa in the ejaculate, the chance of finding them in routine IVF-ICSI is low.

We found that even in Group A (no normal spermatozoa with MSOME), normal spermatozoa were found in 89% of the couples after an extensive search on the day of oocyte pick-up.

When dividing the cohort according to the percentage of normal cells at MSOME, we found a significantly higher pregnancy rate in Group A as compared to Group B or C and as compared to the rest of the cohort (Groups B+C). These results are in accordance with the above explanation: the lower the percentage of normal cells at MSOME the more extensive is the search for high quality spermatozoa, which occurs when IMSI is performed on the day of oocyte pick-up.

A significantly higher pregnancy rate was found in Group B vs. C, as well. Group B can be defined as male subfertility, for which an extensive search for high quality spermatozoa is also conducted; thus, these couples benefit from IMSI also.

The delivery rate in Group A was higher than that in B and C and there was no significant difference between Groups B and C. This further indicates that men with no normal cells at MSOME benefit the most from IMSI.

Based on the assumption that unexplained infertility might be partially explained by covert male factors, O'Neill et al. [4] designed a treatment algorithm based on sperm chromatin integrity to guide the management of couples with apparently unexplained infertility. They found that when these couples failed IUI but had normal sperm DNA fragmentation, IVF resulted in a clinical pregnancy rate of 12.7%. Those with abnormal sperm DNA fragmentation underwent ICSI with ejaculated spermatozoa, resulting in a higher clinical pregnancy rate of 18.7%. Couples with abnormal sperm DNA fragmentation who failed ICSI with ejaculated spermatozoa, achieved a clinical pregnancy rate of 31.0% with surgically retrieved spermatozoa. Despite the high pregnancy rate using ICSI with surgically retrieved spermatozoa, it is an invasive procedure, while MSOME is not.

We found significantly lower total motile count in group A as compared to B and C. Spermatozoa are separated based on density gradient, where morphologically normal spermatozoa have higher densities [16]. Couples with low percentage of normal cells at MSOME, have high percentage of spermatozoa with severe morphological abnormalities; thus, fewer motile spermatozoa are extracted.

According to our results, the younger the female partner, the higher the pregnancy rate. This has been described previously [17]. In addition, transfer of blastocysts resulted in higher pregnancy rates as compared to transferring cleaved embryos. This finding was previously reported in a meta-analysis that showed live birth rate and other outcomes, including pregnancy rate, after fresh IVF-ICSI were significantly improved with blastocyst transfer as compared to cleavage-stage embryo transfer. These results were attributed to better selection of high-quality embryos for implantation [18].

This study is the first to evaluate the role of MSOME and IMSI in treating couples with unexplained infertility. The most significant limitation of the study is the retrospective design.

## Conclusion

Hidden male factor may play a role in the etiology of unexplained infertility. MSOME may be a useful tool for diagnosing hidden male factor and should be considered when evaluating

couples with unexplained infertility who did not conceive after the first cycle. IMSI may provide an advantage to couples with unexplained infertility and none or fewer than 2% normal spermatozoa at MSOME, because the extensive search to find high-quality cells might overcome the hidden male factor which was the obstacle to achieving pregnancy in the previous cycles. Importantly, this method is noninvasive, unlike when spermatozoa are surgically retrieved. Further studies that compare IMSI vs. ICSI among males with severe, isolated teratozoospermia diagnosed by MSOME are needed to strengthen these primary results.

## Supporting information

**S1 Data.**
(XLS)

**S2 Data.**
(XLSX)

## Acknowledgments

The authors express their appreciation to Faye Schreiber for important English language contributions and editing, and to Navah Jelin for contributing to the statistical analysis.

## Author Contributions

**Conceptualization:** Aula Asali, Arie Berkovitz.

**Data curation:** Aula Asali, Netanella Miller, Yael Pasternak, Arie Berkovitz.

**Formal analysis:** Netanella Miller.

**Methodology:** Vita Freger, Michael Belenky, Arie Berkovitz.

**Writing – original draft:** Aula Asali.

**Writing – review & editing:** Aula Asali, Netanella Miller, Yael Pasternak, Vita Freger, Michael Belenky, Arie Berkovitz.

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
