## [Decision Letter · Decision Letter 0]

22 Oct 2019

PONE-D-19-23244

Unexplained infertility treatment, should we change our approach? - An eight-year follow-up study

PLOS ONE

Dear Dr. asali,

Thank you for submitting your manuscript to PLOS ONE. After careful consideration, we feel that it has merit but does not fully meet PLOS ONE’s publication criteria as it currently stands. Therefore, we invite you to submit a revised version of the manuscript that addresses the points raised during the review process.

We would appreciate receiving your revised manuscript by Dec 06 2019 11:59PM. To enhance the reproducibility of your results, we recommend that if applicable you deposit your laboratory protocols in protocols.io, where a protocol can be assigned its own identifier (DOI) such that it can be cited independently in the future. For instructions see: http://journals.plos.org/plosone/s/submission-guidelines#loc-laboratory-protocols

We look forward to receiving your revised manuscript.

Kind regards,

Marco Aurélio Gouveia Alves

Academic Editor

PLOS ONE

**Journal Requirements:**

2. At this time, for purposes of reporting, we ask that you please revise your Materials and Methods section to include the following additional information:

- inclusion and exclusion criteria

- your complete ethics statement

- please specify whether the ethics committee specifically waived the need for informed consent.

:I have read the journal's policy and the author, Dr. Berkovitz, has the following competing interests: Dr. Berkovitz is the Medical Director of the Male Fertility Clinic where the MSOME was performed. The other authors have no conflicts to declare.:

**Comments to the Author**

1. Is the manuscript technically sound, and do the data support the conclusions?

Reviewer #1: Partly

2. Has the statistical analysis been performed appropriately and rigorously? 

Reviewer #1: Yes

3. Have the authors made all data underlying the findings in their manuscript fully available?

Reviewer #1: Yes

4. Is the manuscript presented in an intelligible fashion and written in standard English?

Reviewer #1: Yes

5. Review Comments to the Author

Reviewer #1: This case-control study aims to examine the usefulness of a diagnostic test (MSOME) and therapeutic approach (IMSI), by comparing the outcomes of pregnancy and delivery rates, in a population of couples with unexplained infertility and at least one failed IVF cycle. The cases underwent diagnostic testing and therapeutic approach, the controls did not receive testing and received standard of care. The cases were divided based on MSOME into 3 groups: A (0% normal cells), B (1% normal cells), C (2+% normal cells). When IMSI was used to select the sperm for injection, pregnancy and delivery rates were highest for those groups with the least normal cells by MSOME A>B>C. Controls had pregnancy and delivery rates similar to group C.

Overall, there is a need for additional study in the utility of IMSI, though I would argue, at least at its present level of detail, it’s not clear that this study investigates a population distinct from the male infertility and prior ICSI failure populations that have previously been presented (MSOME and IMSI Reviewed in 2013, PMID 23948449). The study acknowledges the limitations of it being retrospective, with some detail unobtainable. The study does admirably study cumulative pregnancy and delivery rate, including fresh and frozen transfers.

1. Major: Criticism of many unexplained infertility studies stems from differences in inclusion/exclusion criteria for ‘unexplained’ couples. Additional detail in the materials and methods on page 4 (or supplement) regarding the ovulatory, male factor, uterine/tubal/endometriosis, and ovarian reserve workup of the cases and controls would strengthen the ability of the reader to interpret generalizability to their unexplained infertility/mild male factor population. In the discussion it is recognized as a limitation that this level of detail appears unavailable in the control group. But is the case group all unexplained infertility or would some of them qualify as male factor fertility? The novelty of this study is using the technique in an unexplained infertility population rather than male factor infertility, so an ability to differentiate between them is critical to the claim of novelty.

2. Major: studying the pregnancy rate per couple has some advantages, but additional detail regarding the representation of fresh/frozen embryos/numbers of transfers would be appreciated. Additionally, the length of follow-up per couple is inherently different over an overall 8 year timespan. Consider reporting average length of treatment in couples? Do I assume correctly that only the first pregnancy represented by the embryos generated in an IMSI cycle would be represented in the data? Additionally, how was pregnancy defined in your electronic database? Home test, serum?

3. Major: I would presume the comparison of outcomes between controls collected 2010-2018 and cases 2015-2018 would generally find that controls had lower pregnancy/delivery rates, at least with US trends; I don’t know if the same can be extrapolated to other countries. What would a sensitivity analysis comparing the controls from 2015-2018 to cases from 2015-2018 show? On a related question, how was the decision made to include the number of cases you did? Was a power analysis done, or was it a convenience sample based on introduction of an EMR in 2010 and including all cycles that met criteria?

4. Major: The title does not inform the reader of the study design, population, or intervention of this study, and the eight-year follow-up was only for the controls, not the cases. Title revision strongly recommended (consider consulting the STROBE guideline).

5. Minor: I’m not sure I understand/agree with the logic behind the ‘explanation for this apparent contradiction…that a lower percentage of normal cells in MSOME necessitates an extensive search under ultra-magnification to find the highest quality spermatozoa.’ The criteria by which ‘highest quality’ are judged would not seem to be different in group A and B, but the extensiveness of the search has been a major criticism of the IMSI technique and a concern with the time the oocytes wait to be injected.

6. Minor: Bonferroni corrections used for multiple comparisons is great, but it’s not clear what alpha was used to judge significance in the different cases; I like to be able to see how close the p-value is to the alpha.

7. Minor: is the IMSI technique success related to the embryologist/andrologist examining the sperm? I can’t imagine a single embryologist/andrologist performed all the studies, was there a bias in the most experienced performing those for Group A where difficulty in finding normal sperm was anticipated?

6. PLOS authors have the option to publish the peer review history of their article (what does this mean?). If published, this will include your full peer review and any attached files.

Reviewer #1: No

---

## [Author Response · Author response to Decision Letter 0]

18 Dec 2019

1. Major: Criticism of many unexplained infertility studies stems from differences in inclusion/exclusion criteria for ‘unexplained’ couples. Additional detail in the materials and methods on page 4 (or supplement) regarding the ovulatory, male factor, uterine/tubal/endometriosis, and ovarian reserve workup of the cases and controls would strengthen the ability of the reader to interpret generalizability to their unexplained infertility/mild male factor population. In the discussion it is recognized as a limitation that this level of detail appears unavailable in the control group. But is the case group all unexplained infertility or would some of them qualify as male factor fertility? The novelty of this study is using the technique in an unexplained infertility population rather than male factor infertility, so an ability to differentiate between them is critical to the claim of novelty.

Thank you for your comment. 

All the included couples in the case group and control group had unexplained infertility, and had experienced at least three failed IUI cycles and at least one failed IVF-ICSI cycle. Unexplained infertility was an inclusion criterion. The missing data in the control group was not related to the diagnosis of unexplained infertility or the treatment they underwent. 

According to a recent manuscript published in Human Reproduction, subfertility was defined as “couples trying to conceive for at least 12 months. Selected subfertile couples had regular menstrual cycles, at least one patent fallopian tube and a total motile sperm count >1 × 106” (PMID 30395266), which enabled them to try IUI. Therefore, they can be added to the unexplained infertility group. I added the definition of male subfertility to the methods section, page 4. “Male subfertility was defined as a total motile sperm count >1 × 106” [8].”

We emphasized this in the methods section and added exclusion criteria, pages 4,5. 

“Exclusion criteria were Mullerian abnormality or submucosal or subserosal fibroids >7 cm, despite regular menstrual cycles, because of the negative effect of these complications on implantation and pregnancy outcomes [9]; ovulation abnormalities; tubal pathology; and male infertility (when the total motile sperm count was <1 × 106, because IUI treatment does not benefit this group).”

2. Major: studying the pregnancy rate per couple has some advantages, but additional detail regarding the representation of fresh/frozen embryos/numbers of transfers would be appreciated. Additionally, the length of follow-up per couple is inherently different over an overall 8 year timespan. Consider reporting average length of treatment in couples? Do I assume correctly that only the first pregnancy represented by the embryos generated in an IMSI cycle would be represented in the data? Additionally, how was pregnancy defined in your electronic database? Home test, serum?

The pregnancy rate represented in the data is the cumulative pregnancy rate (pregnancy from fresh and frozen embryos). We included the results of the first IMSI only, because we assumed that if there was a hidden male factor it would be resolved in the first IMSI cycle.

I added the definition to the methods section, page 5: “Clinical pregnancy, defined as missed abortion, blighted ovum or fetal pole with heartbeat, was considered a positive result. Chemical pregnancy was considered a negative result”. 

3. Major: I would presume the comparison of outcomes between controls collected 2010-2018 and cases 2015-2018 would generally find that controls had lower pregnancy/delivery rates, at least with US trends; I don’t know if the same can be extrapolated to other countries. What would a sensitivity analysis comparing the controls from 2015-2018 to cases from 2015-2018 show? On a related question, how was the decision made to include the number of cases you did? Was a power analysis done, or was it a convenience sample based on introduction of an EMR in 2010 and including all cycles that met criteria?

I added the following explanation to the statistical analysis section. Page 8.

“Based on the assumption of a 12% difference in pregnancy rates between groups, we calculated that a sample size of 500 couples would be sufficient to provide a power of 80% to show an effect, at two-tailed alpha of 5%.” 

We assumed that the pregnancy rate among couples with unexplained infertility who failed a previous IVF-ICSI and had a hidden male factor would be similar to that of couples with their first treatment, which is described in the literature as 40-45%. The pregnancy rate among the control group is 28%, therefore, the difference we used in calculating the sample size was 12%.

The control group included 285 couples, in which only 61 were in 2015-2018, because of the small sample size we included cases from 2010-2015. The cumulative pregnancy rate among those 61 cases was 24.6%, delivery rate 8.2%, abortion rate 16.39%.

4. Major: The title does not inform the reader of the study design, population, or intervention of this study, and the eight-year follow-up was only for the controls, not the cases. Title revision strongly recommended (consider consulting the STROBE guideline).

Thank you for your comment. We changed the title to: 

 “The possibility of integrating motile sperm organelle morphology examination 

 (MSOME) and intracytoplasmic morphologically-selected sperm injection (IMSI) in 

 treating couples with unexplained infertility: a historical, case-control study”

5. Minor: I’m not sure I understand/agree with the logic behind the ‘explanation for this apparent contradiction…that a lower percentage of normal cells in MSOME necessitates an extensive search under ultra-magnification to find the highest quality spermatozoa.’ The criteria by which ‘highest quality’ are judged would not seem to be different in group A and B, but the extensiveness of the search has been a major criticism of the IMSI technique and a concern with the time the oocytes wait to be injected.

The maximum time from providing the semen sample to oocyte injection is four hours. Couples with higher percentage of normal cells at MSOME do not gain benefit from the extensive search, they have enough normal sperm for fertilization.

In addition, men with higher percentage of normal cells in the ejaculate can be managed with routine IVF ICSI . But, when there are only a few normal spermatozoa in the ejaculate, the chance of finding them in routine IVF ICSI is low. I added this explanation to Page 15.

6. Minor: Bonferroni corrections used for multiple comparisons is great, but it’s not clear what alpha was used to judge significance in the different cases; I like to be able to see how close the p-value is to the alpha.

The alpha used for the Bonferroni Correction was 0.05/3=0.017. In pregnancy rates between groups A and C, p=0.002. In delivery rate between groups A and C, p=0.007, and for groups A and B, p=0.023. For total motile counts between groups A and B p=0.005, between groups A and C p=0.011. I added this to the statistical analysis section, Page 8.

7. Minor: is the IMSI technique success related to the embryologist/andrologist examining the sperm? I can’t imagine a single embryologist/andrologist performed all the studies, was there a bias in the most experienced performing those for Group A where difficulty in finding normal sperm was anticipated?

We have 5 embryologists in our lab. Each one examines the sperm randomly. That means that all 5 embryologists examined each group in the study, without any particular preference.

---

## [Decision Letter · Decision Letter 1]

13 Feb 2020

PONE-D-19-23244R1

The possibility of integrating motile sperm organelle morphology examination (MSOME) with intracytoplasmic morphologically-selected sperm injection (IMSI) when treating couples with unexplained infertility: a historical, case-control study

PLOS ONE

Dear Dr. asali,

Thank you for submitting your manuscript to PLOS ONE. After careful consideration, we feel that it has merit but does not fully meet PLOS ONE’s publication criteria as it currently stands. Therefore, we invite you to submit a revised version of the manuscript that addresses the points raised during the review process.

We would appreciate receiving your revised manuscript by Mar 29 2020 11:59PM. To enhance the reproducibility of your results, we recommend that if applicable you deposit your laboratory protocols in protocols.io, where a protocol can be assigned its own identifier (DOI) such that it can be cited independently in the future. For instructions see: http://journals.plos.org/plosone/s/submission-guidelines#loc-laboratory-protocols

We look forward to receiving your revised manuscript.

Kind regards,

Marco Aurélio Gouveia Alves

Academic Editor

PLOS ONE

Additional Editor Comments (if provided):

The revised version of the manuscript has somewhat improved but there are still several issues to be addressed, particularly concerning the experimental approach.

Reviewers' comments:

Reviewer's Responses to Questions

**Comments to the Author**

1. If the authors have adequately addressed your comments raised in a previous round of review and you feel that this manuscript is now acceptable for publication, you may indicate that here to bypass the “Comments to the Author” section, enter your conflict of interest statement in the “Confidential to Editor” section, and submit your "Accept" recommendation.

Reviewer #1: All comments have been addressed

Reviewer #2: (No Response)

2. Is the manuscript technically sound, and do the data support the conclusions?

Reviewer #1: Yes

Reviewer #2: Partly

3. Has the statistical analysis been performed appropriately and rigorously? 

Reviewer #1: Yes

Reviewer #2: I Don't Know

4. Have the authors made all data underlying the findings in their manuscript fully available?

Reviewer #1: Yes

Reviewer #2: Yes

5. Is the manuscript presented in an intelligible fashion and written in standard English?

Reviewer #1: Yes

Reviewer #2: Yes

6. Review Comments to the Author

Reviewer #1: Comments from prior review have been addressed sufficiently. Limitations of the study design are appropriately acknowledged, details sufficient for the reader to evaluate outcomes.

Reviewer #2: In this manuscript entitled: “The possibility of integrating motile sperm organelle morphology examination (MSOME) with intracytoplasmic morphologically-selected sperm injection (IMSI) when treating couples with unexplained infertility: a historical, case-control study”. Authors asked themselves if the IMSI technique can be used successful on cases of couples with unexplained infertility.

In general, authors don´t explain very well their results and the manuscript is not easy to read. First at all, a general advice for following submission is to include line number. This will not be only beneficial for the reviewers; authors can follow recommendation easier.

One of my concerns is why authors make a difference between MSOME and IMSI. MSOME is not a previous procedure that is part of the process of IMSI? When you read the title and the manuscript it seems that there are 2 different procedures.

Is control group a real control? I am concern about this topic. The only information that we have about them is the reproductive outcome, but we don´t know nothing about the % or normal spermatozoa or the rest of seminogram values. Authors should consider delete this group and built the manuscript discussion making the comparison between groups A, B and C. As they stated: group C is the normal/average group and further comparisons should be done with this group. What we know about the proposed control group?

Authors show beneficial effect of IMSI versus ICSI. This expected result agrees with previous reports that highlight the beneficial outcome of IMSI. Nevertheless, it is hard to understand why the group with higher percentage of anormal spermatozoa had the higher reproductive outcome. The only reason that could explain this contradiction is that the physician made bigger effort to find a normal cell and performed later the ICSI. Once you finish to read the manuscript the conclusion that the reader made is that IMSI is beneficial for those patients with severe morphological sperm problems. But since physician are selecting normal spermatozoa why the rest of group are not improving? Bigger effort should be made to explain this contradictory result.

Minor comments:

Introduction Section:

Please delete this sentence “One study found that ICSI resulted in a similar, cumulative live birth rate as compared with IVF for couples with non-male factor infertility [7]” It doesn´t apport information.

Material and Methods:

Please rephrase it: All couples underwent MSOME when IMSI was done within 6 months of the MSOME. What this means you performed a MSOME analysis and later before 6 months you performed an IMSI. Or maybe did you make a MSOME and you freeze the spermatozoa and later or you performed an IMSI. I am sure this is not the message that you want to send but it is rather confusing. Please clarify.

When you wrote “IVF/ICSI cycles. Male subfertility was defined as a total motile sperm count >1 × 106” What you really mean is less than a million?

Please defined what do you consider “Chemical pregnancy”

When you described the sperm selection “The upper liquid was then removed and the pellet was re-suspended in Quinn's Sperm Washing Medium (Sage In-Vitro Fertilization, Inc.) and centrifuged again for 5 minutes at 600 rpm at room temperature. “What do you mean for the pellet? The spermatozoa that can be found on the bottom of the tube? Do you take the spermatozoa and semen contaminants that can be found on the 40% fraction? Please specify this point.

It is possible to insert in table-3 information about the concentration of the ejaculates. It is the only reason that can explain why different groups that have the same % of motile spermatozoa and same ejaculate volume have different values of total motile count.

Statistical analysis:

Please explain the assumption of a 12% difference in pregnancy rates between groups. How did you calculate this? There is a big gap between the number of couples included in each group. Did the author have this in mind when they performed the statistic? Did you balance statistically this difference?

7. PLOS authors have the option to publish the peer review history of their article (what does this mean?). If published, this will include your full peer review and any attached files.

Reviewer #1: Yes: Jamie Peregrine, MD, MS

Reviewer #2: No

---

## [Author Response · Author response to Decision Letter 1]

24 Feb 2020

Marco Aurélio Gouveia Alves

Academic Editor

PLOS ONE

Re: PONE-D-19-23244R1: The possibility of integrating motile sperm organelle morphology examination (MSOME) with intracytoplasmic morphologically-selected sperm injection (IMSI) when treating couples with unexplained infertility

Dear Professor Alves,

Please find below, our response to the additional comments of reviewer 2. Thank you. 

Reviewer #2

1. One of my concerns is why authors make a difference between MSOME and IMSI. MSOME is not a previous procedure that is part of the process of IMSI? When you read the title and the manuscript it seems that there are 2 different procedures.

Thank you for your comment. MSOME and IMSI are two different procedures. MSOME is an advanced semen analysis that examines the morphology of motile spermatozoa under ultra-magnification of 6300x. It is the process of looking for normal spermatozoa using an inverted light microscope equipped with high-power Nomarski optics, enhanced with digital imaging to achieve magniﬁcation up to 6300×. IMSI is the micromanipulation of the spermatozoa found after extended search and injected into the oocyte.

2. Is control group a real control? I am concern about this topic. The only information that we have about them is the reproductive outcome, but we don´t know nothing about the % or normal spermatozoa or the rest of seminogram values. Authors should consider delete this group and built the manuscript discussion making the comparison between groups A, B and C. As they stated: group C is the normal/average group and further comparisons should be done with this group. What we know about the proposed control group?

Thank you for your comment. According to your recommendation we deleted the control group.

3. Authors show beneficial effect of IMSI versus ICSI. This expected result agrees with previous reports that highlight the beneficial outcome of IMSI. Nevertheless, it is hard to understand why the group with higher percentage of anormal spermatozoa had the higher reproductive outcome. The only reason that could explain this contradiction is that the physician made bigger effort to find a normal cell and performed later the ICSI. Once you finish to read the manuscript the conclusion that the reader made is that IMSI is beneficial for those patients with severe morphological sperm problems. But since physician are selecting normal spermatozoa why the rest of group are not improving? Bigger effort should be made to explain this contradictory result.

You are correct, this concept is difficult to understand. Our cohort included couples who had at least one unsuccessful, conventional IVF-ICSI cycle, assuming that moderate and mild male factor abnormalities can be solved in the first IVF cycle (enough normal spermatozoa in the ejaculate). Failure to achieve pregnancy in the first or second cycle is not due to unresolved male factor and therefore an extended search will not enhance the implantation rate. In contrast, with severe teratozoospermia requires an extended search to be resolved, failure in the previous cycles can be explained by an unresolved, unknown male factor. In support of this assumption, a previous study found an increase in implantation and pregnancy rates using IMSI over conventional ICSI for patients with previously unsuccessful ICSI, with no difference in the results of the first cycle with ICSI vs. IMSI (15. Klement AH, et al. Intracytoplasmic morphologically selected sperm injection versus intracytoplasmic sperm injection: a step toward a clinical algorithm. Fertil Steril 2013;99:1290-3. doi: 10.1016/j.fertnstert.2012.12.020). 

This is explained in page 12, second paragraph.

4. Minor comments:

Introduction Section:

Please delete this sentence “One study found that ICSI resulted in a similar, cumulative live birth rate as compared with IVF for couples with non-male factor infertility [7]” It doesn´t apport information.

We deleted the sentence.

5. Material and Methods:

Please rephrase it: All couples underwent MSOME when IMSI was done within 6 months of the MSOME. What this means you performed a MSOME analysis and later before 6 months you performed an IMSI. Or maybe did you make a MSOME and you freeze the spermatozoa and later or you performed an IMSI. I am sure this is not the message that you want to send but it is rather confusing. Please clarify.

MSOME is the examination of the semen. According to this examination, we divided the cohort into 3 groups according to the percentage of normal spermatozoa in the semen. In our cohort, we included only couples who had MSOME and proceeded to IMSI within 6 months from the MSOME.

Couples who want to proceed to IMSI provide a new semen sample on the day of oocyte pick-up. This was clarified in the Abstract and Methods.

6. When you wrote “IVF/ICSI cycles. Male subfertility was defined as a total motile sperm count >1 × 106” What you really mean is less than a million?

Male subfertility was defined as a total motile sperm count ≥1 × 106.

Male infertility was defined as a total motile sperm count was <1 × 106

7. Please defined what do you consider “Chemical pregnancy”.

When the level of hCG is initially elevated enough to produce a positive result on a pregnancy test but then declines before a gestational sac is detected with ultrasound.

I added this to the Methods section, page 5

8. When you described the sperm selection “The upper liquid was then removed and the pellet was re-suspended in Quinn's Sperm Washing Medium (Sage In-Vitro Fertilization, Inc.) and centrifuged again for 5 minutes at 600 rpm at room temperature. “What do you mean for the pellet? The spermatozoa that can be found on the bottom of the tube? Do you take the spermatozoa and semen contaminants that can be found on the 40% fraction? Please specify this point. 

The pellet is the cellular fraction at the bottom of the tube. Generally, the semen itself will remain above the upper layer and debris and immotile cells will get "stuck" between the upper and lower layers. Most of the motile cells will pass through both layers and reach the bottom of the tube, forming the pellet.

9. It is possible to insert in table-3 information about the concentration of the ejaculates. It is the only reason that can explain why different groups that have the same % of motile spermatozoa and same ejaculate volume have different values of total motile count.

The difference in total motile count is that this parameter was calculated after sample preparation in which spermatozoa are separated based on density gradient, where morphologically normal spermatozoa have higher densities. Couples with low percentage of normal cells at MSOME, have high percentage of spermatozoa with severe morphological abnormalities; thus, fewer motile spermatozoa are extracted. 

This was explained on page 14.

10. Statistical analysis:

Please explain the assumption of a 12% difference in pregnancy rates between groups. How did you calculate this? There is a big gap between the number of couples included in each group. Did the author have this in mind when they performed the statistic? Did you balance statistically this difference?

We did this calculation in order to estimate the sample size of the study group and the control group. Based on your previous comment we deleted the control group.

---

## [Decision Letter · Decision Letter 2]

16 Mar 2020

PONE-D-19-23244R2

The possibility of integrating motile sperm organelle morphology examination (MSOME) with intracytoplasmic morphologically-selected sperm injection (IMSI) when treating couples with unexplained infertility

PLOS ONE

Dear Dr. asali,

Thank you for submitting your manuscript to PLOS ONE. After careful consideration, we feel that it has merit but does not fully meet PLOS ONE’s publication criteria as it currently stands. Therefore, we invite you to submit a revised version of the manuscript that addresses the points raised during the review process.

We would appreciate receiving your revised manuscript by Apr 30 2020 11:59PM. To enhance the reproducibility of your results, we recommend that if applicable you deposit your laboratory protocols in protocols.io, where a protocol can be assigned its own identifier (DOI) such that it can be cited independently in the future. For instructions see: http://journals.plos.org/plosone/s/submission-guidelines#loc-laboratory-protocols

We look forward to receiving your revised manuscript.

Kind regards,

Marco Aurélio Gouveia Alves

Academic Editor

PLOS ONE

Reviewers' comments:

Reviewer's Responses to Questions

**Comments to the Author**

1. If the authors have adequately addressed your comments raised in a previous round of review and you feel that this manuscript is now acceptable for publication, you may indicate that here to bypass the “Comments to the Author” section, enter your conflict of interest statement in the “Confidential to Editor” section, and submit your "Accept" recommendation.

Reviewer #1: (No Response)

Reviewer #2: All comments have been addressed

2. Is the manuscript technically sound, and do the data support the conclusions?

Reviewer #1: Yes

Reviewer #2: Yes

3. Has the statistical analysis been performed appropriately and rigorously? 

Reviewer #1: Yes

Reviewer #2: Yes

4. Have the authors made all data underlying the findings in their manuscript fully available?

Reviewer #1: Yes

Reviewer #2: Yes

5. Is the manuscript presented in an intelligible fashion and written in standard English?

Reviewer #1: Yes

Reviewer #2: Yes

6. Review Comments to the Author

Reviewer #1: I agree with reviewer 2’s recommendation and the authors’ decision to remove the ‘control’ group.

A few other minor comments:

Page 2: ‘2010-2018’. Now that the control group is removed, isn’t the study only 2015-2018?

Page 3: ‘Using this technique together with a micromanipulation system has allowed the introduction of a modified intracytoplasmic sperm injection (ICSI) procedure, known as intracytoplasmic morphologically-selected sperm injection (IMSI)’. Can you say something about ICSI in the second paragraph to introduce this more familiar acronym sooner? Introducing it in the current state is confusing.

Page 4: ‘All couples underwent MSOME when IMSI was within 6 months of the initial MSOME’. This is unclear.

Page 4: ‘at least one failed IVF-ICSI cycle’ vs ‘several, previous failed IVF/ICSI cycles’. To whom was MSOME offered, and who was included in the study? Also if IVF/ICSI and IVF-ICSI the same, recommend consistency.

Page 10: ‘Group A 10 included 55 cases in which no normal cells were found at MSOME, but normal cells were found during IMSI at ovum pick-up for 49 (89.1%). Group B included 184 couples and Group C 32 couples.’ Did all Group B and Group C patients have normal cells found for IMSI? If not, an argument could be made that maybe MSOME is just not predictive of IMSI, even if <6 months, so I would include the information if you have it.

Page 12: consider modifying as bolded: ‘Thus, MSOME can detect subtle organellar malformations in motile spermatozoa that an embryologist might consider normal for fertilization at 200× to 400× magnification [5], especially important when a normal sperm nucleus is a significant factor for successful implantation and pregnancy in ICSI procedures [13,14].

Page 13: the argument is strengthened, but make sure formatting (IVF-ICSI or IVF/ICSI or IVF ICSI) is consistent.

Page 13: It is confusing to have 2 definitions for ‘male subfertility’: both >1 million cells and Group B.

Reviewer #2: Authors adressed correctly all my questions and concerns. Moreover, modifications were performed on the manuscript.

7. PLOS authors have the option to publish the peer review history of their article (what does this mean?). If published, this will include your full peer review and any attached files.

Reviewer #1: Yes: Jamie Peregrine

Reviewer #2: No

---

## [Author Response · Author response to Decision Letter 2]

23 Mar 2020

Response to reviewer-

Page 2: ‘2010-2018’. Now that the control group is removed, isn’t the study only 2015-2018?

Thank you for your comment. You are right. I changed this in the abstract.

Page 3: ‘Using this technique together with a micromanipulation system has allowed the introduction of a modified intracytoplasmic sperm injection (ICSI) procedure, known as intracytoplasmic morphologically-selected sperm injection (IMSI)’. Can you say something about ICSI in the second paragraph to introduce this more familiar acronym sooner? Introducing it in the current state is confusing.

I added a paragraph in page 3 explaining the difference between ICSI and IMSI: "When the spermatozoa with normal morphology and motility selected for ICSI are detected under a magnification of ×400, in IMSI the motile spermatozoa are selected under magnification up to 6300×[5]". 

The advantage of IMSI is that the sperm is detected under magnification up to 6300× which can detect subtle organellar malformations in motile spermatozoa that an embryologist might consider normal for fertilization at 200× to 400× magniﬁcation (ICSI). This explanation is written in page 3 and page 12.

Page 4: ‘All couples underwent MSOME when IMSI was within 6 months of the initial MSOME’. This is unclear.

I changed this sentence to: All couples underwent IMSI within 6 months of the initial MSOME.

Page 4: ‘at least one failed IVF-ICSI cycle’ vs ‘several, previous failed IVF/ICSI cycles’. To whom was MSOME offered, and who was included in the study? Also if IVF/ICSI and IVF-ICSI the same, recommend consistency.

MSOME was offered to couples who had at least one failed IVF-ICSI cycle. I changed this. And changed IVF/ICSI to IVF-ICSI.

Page 10: ‘Group A 10 included 55 cases in which no normal cells were found at MSOME, but normal cells were found during IMSI at ovum pick-up for 49 (89.1%). Group B included 184 couples and Group C 32 couples.’ Did all Group B and Group C patients have normal cells found for IMSI? If not, an argument could be made that maybe MSOME is just not predictive of IMSI, even if <6 months, so I would include the information if you have it.

All the couples in group B and C had normal cells for IMSI. 

Page 12: consider modifying as bolded: ‘Thus, MSOME can detect subtle organellar malformations in motile spermatozoa that an embryologist might consider normal for fertilization at 200× to 400× magnification [5], especially important when a normal sperm nucleus is a significant factor for successful implantation and pregnancy in ICSI procedures [13,14].

I did this.

Page 13: the argument is strengthened, but make sure formatting (IVF-ICSI or IVF/ICSI or IVF ICSI) is consistent.

Thank you. I changed to IVF-ICSI.

Page 13: It is confusing to have 2 definitions for ‘male subfertility’: both >1 million cells and Group B.

All the couples in the cohort had unexplained infertility or male subfertility, when male subfertility is defined as a total motile sperm count ≥1 × 106 (written in page 4), that means that each group may have cases of male subfertility.

---

## [Editor Report · Decision Letter 3]

9 Apr 2020

The possibility of integrating motile sperm organelle morphology examination (MSOME) with intracytoplasmic morphologically-selected sperm injection (IMSI) when treating couples with unexplained infertility

PONE-D-19-23244R3

Dear Dr. asali,

We are pleased to inform you that your manuscript has been judged scientifically suitable for publication and will be formally accepted for publication once it complies with all outstanding technical requirements.

With kind regards,

Marco Aurélio Gouveia Alves

Academic Editor

PLOS ONE
---

## [Editor Report · Acceptance letter]

22 Apr 2020

PONE-D-19-23244R3 

The possibility of integrating motile sperm organelle morphology examination (MSOME) with intracytoplasmic morphologically-selected sperm injection (IMSI) when treating couples with unexplained infertility 

Dear Dr. Asali:

I am pleased to inform you that your manuscript has been deemed suitable for publication in PLOS ONE. Congratulations! Your manuscript is now with our production department. 

With kind regards,

on behalf of

Dr. Marco Aurélio Gouveia Alves 

Academic Editor

PLOS ONE